# Learning by Moving Closer: Adapting Vision Models on Movable Agents without Manual Labeling

## Abstract

One of the most fundamental goals of machine learning is to enable systems to perceive the world. When deploying pre-trained vision models, it is crucial for agents to adapt these models to new environments without relying on human annotations. In this paper, considering movable agents, we advocate a model adaptation framework for *learning to see by moving without manual labeling*. Our approach stems from the following observation: Predictions made by an agent exhibit significant quality variance as the agent moves, e.g., at closer versus farther distances for an object, implying that high-quality predictions can naturally serve as a teacher's output to adapt the model. Since incorrect teacher predictions can mislead adaptation, we develop a unified data sampling-and-weighting (SAW) framework according to prediction confidence, making the loss function an *unbiased* estimator of the clean loss. Experimental results demonstrate that our proposed scheme significantly outperforms prior schemes across various models and datasets.

## 1 Introduction

Vision models are notoriously weak in generalization under distribution shifts, leading to significant performance degradation even with minor variations in data distributions Recht et al. (2018); Koh et al. (2021). As such, model adaptation is essential to enhance perception performance in new scenarios Samuel (1959); Zhu et al. (2023). However, adapting pre-trained vision models, such as object classification models, generally require manual labeling, hindering their ability to autonomously improve over time Ma et al. (2025). For example, an autonomous vehicle navigating a new environment would find it costly, time-consuming, and even impractical to obtain human annotations for adapting the pre-trained vision model. Considering this fact, can the agent still improve its perception capabilities by adapting the pre-trained model, even though there is no label at all?

The problem above falls under the field of test-time adaptation (TTA) Chen et al. (2022); Sun et al. (2020), sometimes called source-free domain adaptation Kundu et al. (2020); Liang et al. (2020). In the vision domain, TTA exploits methods including uncertainty-guided pseudo-labeling Litrico et al. (2023), image/feature generation Zhang et al. (2022), and contrastive learning Sun et al. (2020); Chen et al. (2022). Crucially, the design of TTA systems should 1) operate unsupervised; 2) have no knowledge of pre-training data or pre-training procedure; 3) remain model-agnostic to ensure broad applicability across different architectures, and 4) be insensitive to hyperparameter choices for implementation in varying environments.

In this paper, we present a novel TTA approach for object classification tasks that meets the aforementioned properties. We contend that most of the existing works on TTA cannot satisfy all of these requirements simultaneously, particularly being model-agnostic and hyperparameter-insensitive, because the performance may vary significantly across models and datasets, as we will demonstrate in our experiment section. Besides, compared with prior TTA schemes, our framework provides a *novel* design dimension by exploiting the **prediction variances upon agent moving**. As illustrated in Fig. 1, at the heart of our approach lies the observation that an agent generally yields predictions with significant quality variance upon movement, say, more accurate predictions for closer objects and less accurate predictions for further objects, implying that high-quality predictions can naturally

Figure 1: Illustration of learning by moving. The agent is more likely to produce the correct class, i.e., cat, when moving closer to the object. The high-confidence predictions (on the right) can serve as pseudo labels to guide model adaptation over samples with lower confidence (on the left).

serve as "teacher" output [1] to supervise low-quality predictions without manual labeling. Since the prediction accuracy may not vary strictly monotonically with distance due to potential obstructions, we avoid using naive distance-based metrics and instead use confidence as the metric for selecting teacher output, being consistent with a large body of pseudo-labeling methods Litrico et al. (2023); Zhang et al. (2021). To mitigate the risk of overfitting towards corrupted teacher outputs being selected based on confidence, we propose a two-step sampling-and-weighting (SAW) framework. In the first step, a data sampling strategy is proposed to select "student" and "teacher" data samples by comparing the prediction confidence of the object from different observations for label propagation. In the second step, we assign importance weightings to the selected "student" data samples by reflecting on how likely the teacher's predictions are incorrect, ensuring the corresponding loss function an unbiased estimator of the clean loss.

Our contributions are summarized as follows.

- We present the idea of learning to see by movement, which propagates the knowledge of high-quality predictions to low-quality predictions for a movable agent. Due to its simplicity, the framework is model-agnostic and hyperparameter-insensitive – properties not achieved by most TTA approaches.

- Focusing on image classification tasks, we propose the SAW framework to select data samples and assign importance weightings. We theoretically show that this framework leads to unbiased estimator of the clean loss under the presence of corrupted teacher predictions.

- We conduct extensive experiments on object classification tasks to 1) demonstrate the effectiveness of learning by moving and 2) show the significant improvement brought by the proposed SAW framework compared with other benchmarks.

## 2 RELATED WORK

### 2.1 TEST-TIME ADAPTATION

TTA adapts models to unlabeled test data without requiring access to the source datasetChen et al. (2022); Litrico et al. (2023); Yuan et al. (2023); Karim et al. (2023); Chen et al. (2024); Joo & Klabjan (2024). In Sun et al. (2020), test-time training introduces a self-supervised auxiliary rotation prediction task to be optimized jointly during both source and target training. However, this approach requires modification to the pre-training, limiting its applicability. In Chen et al. (2022), Chen *et al.* propose to refine predictions within a self-supervised strategy. However, their work does not take into account the noise inherent in pseudo-labels, resulting in detrimental noise overfitting and thus poor adaptation. In Litrico et al. (2023), Litrico *et al.* propose a framework that reweights classification loss according to pseudo-label uncertainty to improve the robustness against the noise by progressively refining the pseudo-labels via neighborhood knowledge aggregation. In Gong et al. (2022), by observing that data can be temporally correlated in application scenarios, e.g., autonomous driving, Gong *et al.* propose a scheme against such non-IID data streams with time correlation. However, while Gong *et al.* consider the time correlation of data frames upon

---

[1] In this paper, we call high-quality predictions used for supervising model training as teacher output and the training data samples guided by teacher output as "student" data samples.

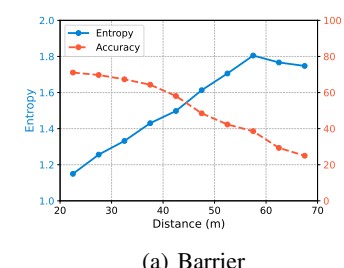 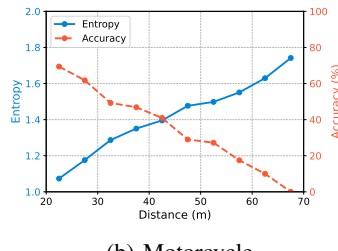

(a) Barrier           (b) Motorcycle

Figure 2: Average entropy and classification accuracy versus the distance for two classes of objects, i.e., barriers and motorcycles, in the nuScenes dataset Caesar et al. (2020) with a pre-trained ResNet-50 model. Accuracy and entropy (confidence) **vary significantly as agents move**.

agent movement, their goal is to combat its negative effects instead of exploit the agent movement for adaptation.

As mentioned earlier, most existing TTA approaches cannot satisfy all the required properties, especially insensitivity to hyperparameters Boudiaf et al. (2022). Despite the shared goal of adapting models to unlabeled datasets, our framework stands out by not only satisfying all the requirements but also exploiting a new dimension of design, i.e., cross-frame prediction variance induced by agent movement.

### 2.2 OTHER UNSUPERVISED LEARNING APPROACHES

Apart from TTA, there are some other unsupervised learning frameworks, yet they are designed for different goals. **Contrastive learning** is a powerful unsupervised learning paradigm that aims to learn an embedding space in which similar samples stay close and dissimilar ones stay apart. A seminal work in this field is InfoNCE van den Oord et al. (2019), which introduces a contrastive loss function to compare positive samples against multiple negatives. In He et al. (2019), He *et al.* propose Momentum Contrast (MoCo) for unsupervised visual representation learning by building a dynamic dictionary with a queue and a moving-averaged encoder Bachman et al. (2019). In Agrawal et al. (2015), Agrawal *et al.* propose a feature learning approach based on egomotion information of agents. Despite being unsupervised, contrastive learning aims to learn transferable features during the *pre-training stage*, which are then applied to specific downstream tasks with labeled data samples. This fundamentally differs from our problem at hand, which targets the model adaptation (fine-tuning) stage.

While most **knowledge distillation** (KD) schemes focus on labeled settings Hinton (2015), KD can naturally be applied to unlabeled datasets by employing a powerful teacher model to generate pseudo labels for a student model Park et al. (2019). In Iliopoulos et al. (2022), a "debiasing" reweighting framework is proposed to reweight the loss function of a student model in an unlabeled dataset. In Kontonis et al. (2024), a student-label mixing (SLaM) approach is proposed to modify the student's loss by considering the noise model of the teacher. However, our approach differs from these label-free KD mechanisms by exploiting the prediction variation upon agent moving to our benefit, where both "teacher" and "student" predictions *come out of the same model*, thus eliminating the need for a powerful teacher model.

Finally, as a remark, since we are interested in *unsupervised* settings, the large body of works on **semi-supervised learning** Chapelle et al. (2009); Tarvainen & Valpola (2018); Laine & Aila (2017); Wang et al. (2023), which trains a model with both labeled and unlabeled data samples, is not included in our discussions herein.

### 3 MODEL ADAPTATION FRAMEWORK

In this section, we present our proposed unified SAW framework. In subsection 3.1, we first review the considered multiclass classification problem. In subsection 3.2, we introduce the proposed SAW framework.

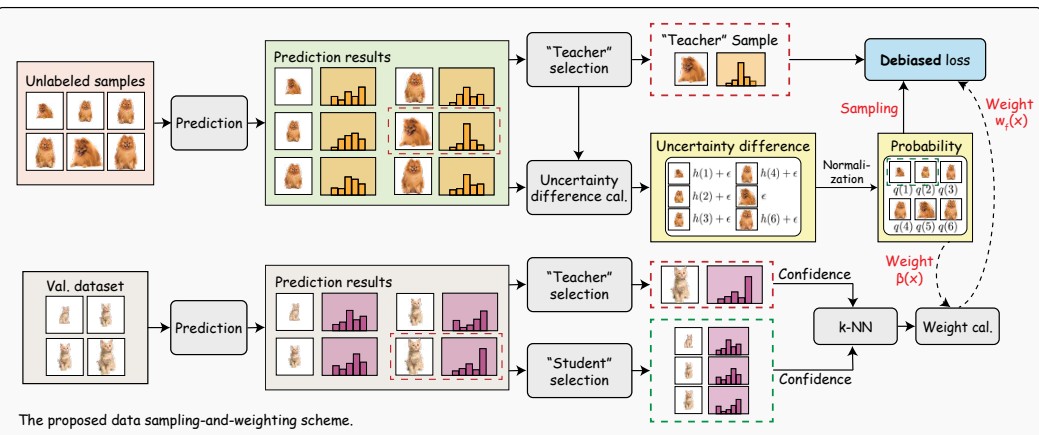

(a) The data sampling-only scheme in Section 3.2.1.

(b) The data sampling-and-weighting scheme in Section 3.2.1 and 3.2.2.

Figure 3: Overview of the proposed methods. (a) achieves unbiased loss only if predictions on the teacher samples are ground-truth; To address this issue, (b) achieves unbiased loss by employing an additional validation dataset. Samples enclosed within the red dashed box correspond to the teacher samples, whereas those within the green dashed box correspond to student samples.

## 3.1 MULTICLASS CLASSIFICATION

We consider a classical problem of multiclass classification, where the objective is to map input data (an image or part of an image) to one of several possible categories. Let $\mathcal{X}$ denote the input space and $\mathcal{Y} = \{1, \ldots, C\}$ represent the label space with $C$ distinct classes. Our training dataset $\mathcal{S} = \{(x_1, y_1), (x_2, y_2), \ldots, (x_n, y_n)\}$ consists of $n$ samples drawn from the distribution $\Pi$, where $\Pi$ is an unknown distribution over $\mathcal{X}$ and labels $\mathcal{Y}$. Each sample $(x_i, y_i)$ pairs an input $x_i \in \mathcal{X}$ with its corresponding label $y_i \in \mathcal{Y}$. The task is to learn a function $f : \mathcal{X} \rightarrow \mathcal{Y}$ that minimizes the risk associated with misclassification. To measure the performance of $f$, we define the expected risk

$$R(f) = \mathbb{E}[\ell(y, f(x))], \tag{1}$$

where $(x, y)$ is sampled from $\Pi$ and $\ell : \mathbb{R}^C \times \mathbb{R}^C \rightarrow \mathbb{R}_+$ is a loss function with $\ell(y, f(x))$ representing the risk incurred for prediction $f(x)$ when the ground-truth label is $y$. We aim to find the function $f^\star$ that minimizes the empirical risk

$$f^\star = \arg\min_{f \in \mathcal{F}} \frac{1}{n} \sum_{(x,y) \in S} \ell(y, f(x)), \tag{2}$$

where $\mathcal{F}$ is a predefined hypothesis class of predictors, often parameterized by their weights.

However, as can be seen, the above classical optimization demands ground-truth label $y$, which is assumed to be unavailable in our case. In what follows, we aim to propose the SAW framework to optimize $f$ effectively with a debiasing reweighting loss function without any ground-truth labels by exploiting the mobility of agents.

## 3.2 DATA SAMPLING-AND-WEIGHTING FRAMEWORK

As an agent moves, it can capture a sequence of images of the same object. We can assign pseudo-labels produced from accurate object predictions to inaccurate data samples, thus guiding the adaptation process. We illustrate how to achieve this as follows.

### 3.2.1 DATA SAMPLING SCHEME

**Uncertainty-aware data sampling.** We first employ the uncertainty-aware sampling principle to select a subset of unlabeled samples for fine-tuning. With a significant variance in prediction quality upon agent movement (shown in Fig. 2), our method uses the sample with the least uncertainty to supervise the sample with high uncertainty. Without loss of generality, we use entropy as the metric to measure uncertainty, as it is widely used in unsupervised/semi-supervised learning methods due to its high correlation with accuracy Litrico et al. (2023); Zhang et al. (2021)[2]. We select the sample with the least uncertainty in each sequence and record the output as "teacher" predictions, where a sequence corresponds to a set of images containing the same object[3]. With teacher predictions, we calculate *entropy difference* $h(x)$, defined as the difference between the entropy of sample $x$ and its corresponding teacher prediction $y^t(x)$ in the sequence. We assign $h(x) + \epsilon$ to each training sample and compute the sampling probability $q(x)$ as the normalized value of $h(x) + \epsilon$ across the training dataset, where $\epsilon$ is a small value that prevents zero sampling probability for teacher samples. After obtaining distribution $q(x)$, we select $m$ data samples from the dataset according to $q(x)$, resulting in the selected dataset $\mathcal{X}_m$.

**Debiasing the sampling process.** However, the sampling process above introduces potential bias, as the resulting selected samples, drawn from an instrumental distribution $\mathbb{Q}$ according to the importance sampling principle, may not perfectly match the true data distribution $\Pi$. If teacher output is perfect, this empirical risk can be debiased by data weighting as follows

$$\hat{R}_q = \frac{1}{n} \sum_{x \in \mathcal{X}} \beta(x) \ell \left( y^t(x), f(x) \right), \tag{3}$$

where $\beta(x)$ is the importance weight for sample $x$ to correct the mismatch between distribution $\Pi$ and instrumental distribution $\mathbb{Q}$.

Assuming $y^t(x)$ is the ground-truth label, i.e., $y^t(x) = y$, we have the following proposition.

**Proposition 3.1.** *If $\beta(x) = \frac{\pi(x)}{q(x)}$, where $\pi(x)$ is the probability density of the distribution $\Pi$ and $q(x)$ is the probability density of the instrumental distribution $\mathbb{Q}$, then $\hat{R}_q$ in equation 3 is an unbiased estimator of $R(f)$ assuming $y^t(x) = y$.*

Note that $\beta(x) \propto \frac{1}{q(x)}$ if $x$ is i.i.d. from $\Pi$. Proposition 3.1 is a direct consequence of the definition of $\beta(x)$, as shown in Kanamori & Shimodaira (2003). The overview of the proposed data sampling framework is illustrated in Fig. 3(a).

### 3.2.2 DATA WEIGHTING SCHEME

The data sampling scheme above, together with its debiasing process, makes $\hat{R}_q$ an unbiased estimator of $R(f)$ only if the teacher prediction is true, i.e., $y^t(x) = y$. In practice, teacher predictions can be inaccurate, which can harmfully affect the learning process, resulting in overfitting toward corrupted labels. To mitigate this issue, we further present a data weighting scheme by adapting the weighting method in Iliopoulos et al. (2022).

Let us consider a noise model for teacher predictions as follows. Let $\mathbb{X}$ be an unknown distribution over objects, i.e., $x \sim \mathbb{X}$. We assume the existence of a ground-truth classifier so that each $x \in \mathcal{X}_m$ is associated with the ground-truth label $y \in \mathcal{Y} = \{1, 2, \ldots, C\}$. Therefore, a clean labeled objects from $x \in \mathcal{X}_m$ follows $(x, y) \sim \tilde{\mathbb{Q}}_{\text{clean}}$. To quantify the noise induced by teacher predictions, we

---

[2]We could use any metric (not necessarily entropy) that correlates well with the accuracy of the corresponding models. We also employ margin score Roth & Small (2006) as the confidence metric in our simulations.

[3]In practice, this sequence can be selected based on time correlation information and class-agnostic object tracking methods.

consider a stochastic teacher that outputs a "corrupted" label $f_{\text{false}}(x)$ with probability $p(x)$ and the ground-truth label $y$ with probability $1 - p(x)$. Let $\tilde{\mathbb{Q}}_t$ denote the induced distribution over data samples and pseudo labels produced by teacher predictions.

Clearly, empirical risk $R^q(f)$ with respect to a predictor $f$ and samples from the induced distribution $\tilde{\mathbb{Q}}_t$ is a biased estimator of risk $R(f)$ in (1), as formalized below.

**Proposition 3.2.** *By defining the biased component $B(f) = \mathbb{E}_{x \sim \mathbb{X}}\left[p(x) \cdot (d_f(x) - 1) \cdot \ell\left(y, f(x)\right)\right]$, we have:*

$$\mathbb{E}\left[R^q(f)\right] = R(f) + B(f), \tag{4}$$

*where*

$$d_f(x) = \frac{\ell\left(f_{\text{false}}(x), f(x)\right)}{\ell\left(y, f(x)\right)}. \tag{5}$$

The proof can be found in Appendix A.1. Notice that, as expected, the bias of the unweighted predictor is a function of the capabilities of the teacher, which depends on how often they corrupt the label and the "distortion" this corruption causes.

To address the bias introduced by the corrupted labels, for each $x \in \mathcal{X}$, we assign the weight

$$w_f(x) = \frac{\beta(x)}{1 + p(x)\left(d_f(x) - 1\right)}. \tag{6}$$

The weighted empirical risk, therefore, can be expressed as

$$R^w(f) = \frac{1}{m} \sum_{x \in \mathcal{X}_m} w_f(x)\ell(y^t(x), f(x)). \tag{7}$$

As shown below, the weighted empirical risk in equation 7 is an unbiased estimation of $R(f)$.

**Proposition 3.3.** *(Unbiased Estimator) Assume that $\mathcal{X}_m$ is drawn from the induced distribution $\tilde{\mathbb{Q}}_t$, then we have:*

$$\mathbb{E}\left[R^w(f)\right] = R(f). \tag{8}$$

The proof can be found in Appendix A.2.

Given a sufficiently large dataset sampled from $\tilde{\mathbb{Q}}_t$, optimizing an unbiased estimator for the risk should provide a better approximation for $\min_{f \in \mathcal{F}} R(f)$ than optimizing an estimator with constant bias. From equation 5 and equation 6, we can observe that the weight for each sample $x$ depends on two factors: (i) the likelihood that the teacher corrupts its label, and (ii) the extent to which this corrupted label distorts the risk. Specifically, if the probability of a corrupted label is high, meaning the teacher's prediction is very likely to be incorrect, we should assign a smaller weight and learn less from the teacher. Similarly, if the distortion term is large, we should also assign a smaller weight to minimize the influence of the teacher's erroneous prediction.

**Estimating (6).** To calculate the weights in (6), one hurdle is that $p(x)$ and $d_f(x)$ are usually unknown. To address this issue, we assume the existence of a validation dataset $\mathcal{S}_v = \{(x_1, y_1), (x_2, y_2), \ldots, (x_v, y_v)\}$, which can be a public dataset. The validation set $\mathcal{S}_v$ is divided into two parts based on the confidence scores of images from a pre-trained model: the image with the highest confidence score in each sequence is designated as a teacher image, while the remaining images are designated as student images. With the validation dataset, we estimate the weights via the Nearest Neighbors (NN) method. Due to the page limitation, the detailed process of estimating the weights is outlined in Algorithm 1 in Appendix A.4.

The SAW framework is illustrated in Fig. 3(b) and summarized in Algorithm 2 in Appendix A.4.

### 3.2.3 EXTENSION TO CLASSIFICATION IN OBJECT DETECTION MODELS

The main body of this work focuses on single-label classification, serving as a proof-of-concept study for learning by moving, similar to most TTA schemes Chen et al. (2022); Litrico et al. (2023); Gong et al. (2022). In practice, modern vehicles, robots, or drones often implement object detection models, such as Faster R-CNN, to detect and classify *multiple* objects in a single image. We discuss how to extend our framework to accommodate this case in Appendix A.11.

## 4 EXPERIMENTAL RESULTS

In this section, we present our experimental results. Due to page limit, we highlight the key observations and results in this section and relegate some experimental results and discussions to the supplementary materials.

### 4.1 EXPERIMENTAL SETTINGS

We conduct experiments using autonomous driving datasets nuScenes Caesar et al. (2020) and KITTI Geiger et al. (2013), as well as embodied agent datasets (selected from JRDB Martin-Martin et al. (2021), MOT Voigtlaender et al. (2019), and LaSOT Fan et al. (2019)). Since the datasets are captured by movable agents, they naturally provide images of the same object from varying distances and with different levels of confidence and accuracy. Moreover, we construct instance sequences by extracting specific objects from images.

We adopt two widely-used model architectures for evaluation, i.e., a convolutional neural network, ResNet-50, and a Transformer, Swin-T Liu et al. (2022). We also demonstrate that our method can be extended to adapt the classification part of Faster R-CNN, a well-known object detection model, in Appendix A.12. In our experiments, the model is trained using the cross-entropy loss function. For ResNet-50, the learning rate is set $10^{-5}$ for adaptation and $10^{-4}$ for pre-training, a momentum coefficient of 0.9, and a weight decay of $5 \times 10^{-4}$. The Swin-T configuration was adopted with window size 7 and patch size 4. Base learning rate is set $10^{-5}$ with 5 epoch warmup.

For all experiments, we pre-train these models in the Singapore Onenorth scenario on the nuScenes dataset. We test i) **cross-region** adaptation (from Singapore to Boston), which adapts the pretrained model in the Boston Seaport scenario on the nuScene dataset, ii) **cross-dataset** adaptation, which adapts the pre-trained model on the KITTI dataset, and iii) **cross-domain** adaptation, which adapts the pre-trained model on the embodied agent dataset. The latter two are provided in supplementary materials.

### 4.2 COMPARISON WITH BENCHMARKS

We compare our data sampling-only (**S-only**) (Fig. 3(a)) and **SAW** (Fig. 3(b)) schemes with the following benchmarks. **AdaCon** Chen et al. (2022), **PLUE** Litrico et al. (2023), **RoTTA** Yuan et al. (2023), **C-SFDA** Karim et al. (2023), **UPA** Chen et al. (2024), **AnCon** Joo & Klabjan (2024), and **FSL:** fully-supervised learning (showing the upper bound). We also include two additional settings for the ablation study: **W-only:** only weighting scheme in Section 3.2.2, and **w/o SAW**: uniform weightings and random sampling. Since the aforementioned TTA benchmarks are released with ResNet-based implementations, their default hyperparameter settings yield relatively poor performance when directly applied to transformers, revealing their sensitivity to hyperparameters across different model architectures. We therefore reimplement them on Swin-T with appropriate hyperparameter choice. Moreover, RoTTA employs a robust batch normalization to estimate normalization statistics, which is not well-suited for Swin-T. Therefore, we do not evaluate it on Swin-T.

In Table 1, we evaluate our proposed framework and benchmarks under both Swin-T and ResNet-50 models in the nuScenes dataset. As observed, the proposed SAW framework consistently outperforms other benchmarks significantly. Besides, S-only performs only slightly worse than the SAW framework, yet still surpasses all other baselines across both models. This phenomenon indicates that our approach can achieve superior performance even with the S-only scheme without requiring a validation dataset.

In addition, we have included **per-class accuracy** to further demonstrate the effectiveness of our framework across diverse object categories. As shown in Table 2, our framework substantially outperforms pre-trained models in terms of per-class accuracy - for ResNet-50 with 8 classes improved and only 1 class degraded (1 class remains the same) and for Swin-T with all 10 classes improved. This is because SAW is designed to mitigate the impact of noisy pseudo-labels: Our Proposition A.1 shows that when the probability of pseudo-label corruption is high and the distortion is large, a smaller weighting factor $w_f(x)$ shall be assigned to the sample. The simulation results demonstrate that our framework outperforms other baselines substantially in combating per-class biases.

Table 1: Performance in the Boston Seaport scenario on the nuScenes dataset. Test accuracy after 100 epochs under varying pre-trained model accuracy for this scenario (i.e., 75%, 70%, and 65%). The value in the bracket represents the standard deviation. Boldface highlights the highest accuracy improvement for each pre-trained model, except for the fully-supervised learning benchmark. ↑ and ↓ indicate performance improvement/degradation compared with the pre-trained model, respectively.

| Pre-trained | ResNet-50 | | | Swin-T | | |
|---|---|---|---|---|---|---|
| | 75% | 70% | 65% | 75% | 70% | 65% |
| w/o SAW | 76.18↑(± 0.28) | 72.05↑(± 0.28) | 62.17↓(± 0.37) | 76.69↑(± 0.26) | 72.55↑(± 0.25) | 40.23↓(± 0.43) |
| AdaCon | 77.88↑(± 0.19) | 73.63↑(± 0.15) | 66.51↑(± 0.18) | 76.85↑(± 0.33) | 72.83↑(± 0.47) | 62.28↓(± 0.61) |
| PLUE | 77.94↑(± 0.12) | 73.82↑(± 0.21) | 67.28↑(± 0.29) | 77.43↑(± 0.24) | 73.55↑(± 0.30) | 59.91↓(± 0.35) |
| RoTTA | 78.67↑(± 0.21) | 75.13↑(± 0.33) | 68.20↑(± 0.31) | — | — | — |
| C-SFDA | 78.71↑(± 0.22) | 75.37↑(± 0.26) | 68.34↑(± 0.34) | 77.75↑(± 0.18) | 72.20↑(± 0.19) | 66.43↑(± 0.25) |
| UPA | 76.87↑(± 0.36) | 74.13↑(± 0.34) | 57.60↓(± 0.48) | 78.24↑(± 0.26) | 73.15↑(± 0.23) | 68.12↑(± 0.28) |
| AnCon | 79.23↑(± 0.29) | 75.61↑(± 0.36) | 68.23↑(± 0.38) | 80.38↑(± 0.27) | 76.28↑(± 0.30) | 70.59↑(± 0.32) |
| W-only (ent) | 77.22↑(± 0.22) | 73.63↑(± 0.21) | 65.43↑(± 0.17) | 77.82↑(± 0.23) | 74.83↑(± 0.12) | 43.96↓(± 0.35) |
| S-only (ent) | 80.76↑(± 0.23) | 76.78↑(± 0.26) | 69.16↑(± 0.21) | 89.48↑(± 0.11) | 89.23↑(± 0.13) | 89.25↑(± 0.12) |
| **SAW (ent)** | **81.37↑(± 0.15)** | **78.07↑(± 0.17)** | 69.87↑(± 0.23) | **90.25↑(± 0.15)** | 90.18↑(± 0.08) | 90.09↑(± 0.07) |
| W-only (ms) | 77.23↑(± 0.11) | 73.41↑(± 0.34) | 65.36↑(± 0.25) | 77.80↑(± 0.18) | 74.85↑(± 0.15) | 43.89↓(± 0.30) |
| S-only (ms) | 80.69↑(± 0.24) | 76.68↑(± 0.30) | 69.48↑(± 0.26) | 89.50↑(± 0.22) | 89.10↑(± 0.32) | 89.01↑(± 0.41) |
| **SAW (ms)** | 81.21↑(± 0.25) | 77.91↑(± 0.16) | **70.02↑(± 0.18)** | 90.22↑(± 0.16) | **90.21↑(± 0.08)** | **90.04↑(± 0.23)** |
| FSL | 88.66↑(± 0.12) | 87.83↑(± 0.18) | 86.87↑(± 0.25) | 92.30↑(± 0.06) | 92.12↑(± 0.15) | 91.98↑(± 0.28) |

## 4.3 IMPACT OF VARIOUS KEY FACTORS

**Effect of hyperparameters.** For hyperparameters, we keep the learning rate the same for all baselines. For the S-only scheme, there is no other learning hyperparameter to fine-tune across datasets and models. For the SAW scheme, we show the influence of the neighborhood size $k$ in the $k$-NN method in Table 3 in Appendix A.6. We observe that our framework maintains very stable performance across different $k$. The phenomenon shows that both S-only and SAW are insensitive to hyperparameters. This insensitivity stems from our intuitive design, which does not rely on modifications to the learning objective.

**Effect of the number of selected samples.** Table 4 in Appendix A.7 illustrates test accuracy by varying the number of training samples. Under various $m$, our SAW framework substantially outperforms other benchmarks and the pre-trained model accuracy, demonstrating the robustness of our approaches under varied experimental settings.

**Effect of weighting.** As can be seen in Table 1, sampling-only scheme itself achieves remarkable performance. However, weighting is still useful in the SAW framework, offering consistent improvement over the S-only counterpart, because it leads to an unbiased estimate of the risk.

**Effect of pre-trained model accuracy.** In Table 1, we also observe that adaptation performance improvement/decline is greatly influenced by pre-trained model accuracy. A lower-accuracy model produces more corrupted pseudo labels, which thereby misguide the adaptation process and even cause severe degradation, for example, with Swin-T (65% pre-trained accuracy), "W-only" and "w/o SAW" drop accuracy by over 20%. However, our SAW framework demonstrates consistent and remarkable performance improvement even in such challenging scenarios.

**Effect of confidence metrics.** While we use entropy as the confidence metric for data sampling and weighting, Table 1 shows that using either entropy (ent) or marginal score (ms) as the confidence metric achieves comparable performance, suggesting that one may adopt either of them.

**Effect of sequence construction.** We construct instance sequence using the ground truth bounding boxes in the dataset. Indeed, our framework only requires a weak assumption that the same objects can be associated across frames, which can be achieved by class-agnostic tracking based on off-the-shelf object tracking techniques. To validate our argument, we provide simulations to show the impact of tracking algorithms in Table 8 in Appendix A.10. The method "Well Constructed" implies training on perfect sequence construction (based on object ID in the dataset), whereas the method "Based on Tracking" indicates the sequence constructed based on an object tracking approach Qin et al. (2024). It is shown that the latter suffices to construct sequences, which only introduces a very negligible performance degradation. This will demonstrate the practicality of the proposed framework.

Table 2: Per-class test accuracy in the Boston Seaport scenario on the nuScenes dataset (Pre-trained model accuracy for the Boston scenario: 70%). $\rightarrow$ represents that the performance remains the same with the pre-trained model. The confidence metric used for data sampling and weighting is entropy.

| | ResNet-50 | | | | | | | | | |
|---|---|---|---|---|---|---|---|---|---|---|
| Class | Car | Ped | Cone | Trailer | Bus | Moto | Barrier | Const-Veh | Bicycle | Truck |
| Pre-trained | 82.42 | 77.35 | 70.46 | 77.22 | 62.86 | 61.11 | 53.09 | 49.64 | 48.15 | 23.28 |
| w/o SAW | 87.41↑ | 81.05↑ | 76.95↑ | 75.32↓ | 61.90↓ | 47.22↓ | 52.31↓ | 49.68↑ | 46.91↓ | 18.84↓ |
| AdaCon | 88.38↑ | 85.29↑ | 77.42↑ | 81.00↑ | 68.10↑ | 50.05↓ | 53.09→ | 40.53↓ | 37.50↓ | 14.64↓ |
| PLUE | 88.79↑ | 84.98↑ | 75.55↑ | 78.67↑ | 64.87↑ | 55.89↓ | 54.24↑ | 46.15↓ | 39.71↓ | 15.51↓ |
| RoTTA | 87.41↑ | 83.78↑ | 73.12↑ | 74.74↓ | 65.39↑ | 52.53↓ | 58.33↑ | 40.53↓ | 37.62↓ | 12.87↓ |
| C-SFDA | 87.96↑ | 84.05↑ | 78.24↑ | 72.50↓ | 62.86→ | 62.58↑ | 54.73↑ | 51.16↓ | 35.67↓ | 18.50↓ |
| UPA | 86.43↑ | 83.02↑ | 77.84↑ | 74.97↓ | 69.52↑ | 57.67↓ | 48.84↓ | 45.86↓ | 28.88↓ | 16.73↓ |
| AnCon | 88.91↑ | 86.31↑ | 81.42↑ | 78.67↑ | 68.10↑ | 51.48↓ | 54.24↑ | 47.22↓ | 42.54↓ | 16.73↓ |
| W-only | 87.83↑ | 83.04↑ | 79.58↑ | 75.95↓ | 66.67↑ | 52.78↓ | 53.09→ | 51.80↑ | 48.27↑ | 20.06↓ |
| S-only | 90.37↑ | 88.72↑ | 83.89↑ | 76.58↓ | 71.43↑ | 52.78↓ | 54.24↑ | 50.36↑ | 53.09↑ | 19.14↓ |
| **SAW** | 91.21↑ | 89.83↑ | 83.89↑ | 77.85↑ | 73.33↑ | 61.11→ | 56.81↑ | 56.12↑ | 54.32↑ | 20.67↓ |
| | Swin-T | | | | | | | | | |
| Class | Car | Ped | Cone | Trailer | Bus | Moto | Barrier | Const-veh | Bicycle | Truck |
| Pre-trained | 85.45 | 73.42 | 74.68 | 72.78 | 60.95 | 65.56 | 52.25 | 51.76 | 49.38 | 24.37 |
| w/o SAW | 87.41↑ | 78.24↑ | 76.05↑ | 76.53↑ | 70.47↑ | 75.56↑ | 56.32↑ | 49.92↓ | 46.58↓ | 19.14↓ |
| AdaCon | 89.72↑ | 80.02↑ | 78.10↑ | 73.90↑ | 66.83↑ | 73.92↑ | 50.37↓ | 45.96↓ | 48.65↓ | 15.13↓ |
| PLUE | 90.08↑ | 78.10↑ | 78.26↑ | 76.53↑ | 64.43↑ | 68.37↑ | 54.18↑ | 47.31↓ | 49.19↓ | 16.83↓ |
| C-SFDA | 90.30↑ | 81.23↑ | 77.41↑ | 74.55↑ | 65.95↑ | 66.83↑ | 55.83↑ | 45.66↓ | 36.59↓ | 18.96↓ |
| UPA | 88.90↑ | 80.20↑ | 80.61↑ | 71.65↓ | 69.48↑ | 58.23↓ | 58.59↑ | 46.46↓ | 38.90↓ | 17.27↓ |
| AnCon | 92.80↑ | 84.81↑ | 82.47↑ | 75.83↑ | 67.16↑ | 62.64↓ | 52.68↑ | 43.26↓ | 39.18↓ | 17.36↓ |
| W-only | 87.27↑ | 76.88↑ | 78.89↑ | 75.10↑ | 72.05↑ | 75.82↑ | 63.35↑ | 58.46↑ | 48.05↑ | 22.39↓ |
| S-only | 93.69↑ | 91.71↑ | 94.10↑ | 90.80↑ | 90.65↑ | 87.11↑ | 85.39↑ | 86.14↑ | 75.24↑ | 35.53↑ |
| **SAW** | 95.27↑ | 92.76↑ | 94.00↑ | 92.85↑ | 90.19↑ | 88.29↑ | 86.95↑ | 87.22↑ | 76.19↑ | 40.44↑ |

## 4.4 CROSS-DATASET AND CROSS-DOMAIN EVALUATION

We also conduct the simulations on the KITTI dataset and an embodied agent dataset to show the superiority of our proposed schemes. Due to page limit, the results on the KITTI dataset are presented in Appendix A.8, and the experiments on the embodied agent domain are provided in Table 7 in Appendix 4.4.

## 4.5 ADAPTING THE CLASSIFICATION MODULE OF OBJECT DETECTION MODELS

As mentioned in Section 3.2.3, our proposed SAW framework can also be applied to the classification module of object detection models. We use a well-known object detection model, Faster R-CNN Ren et al. (2015); Wu et al. (2019), as an example by first pretraining the model on the Singapore Onenorth scenario and then adapting only its classification network toward the Boston Seaport scenario based on the SAW framework. The results show that our SAW framework greatly improves the classification performance of Faster R-CNN compared with pre-trained models. The results are provided in Appendix A.12.

## 5 CONCLUSION AND LIMITATION

In this paper, we have proposed a general framework for movable agents to adapt vision models without manual labeling. Our idea is to exploit the variance in prediction quality for the same object upon agent movement. Since incorrect close object predictions can mislead model adaptation, we have developed a unified data sampling-and-weighting framework to judiciously sample training data and assign weightings. We have theoretically shown that the framework leads to unbiased learning with respect to the loss function. Experimental results have demonstrated that our proposed adaptation method can significantly improve model accuracy without manual labeling in object classification tasks and outperform existing test-time adaptation schemes.

This work serves as a proof-of-concept study for "learning by moving", focusing on object classification. However, it is promising to extend this idea to other vision tasks, such as object detection and segmentation, because agent movement can provide valuable information for model adaptation in these tasks as well. These aspects can be exploited in future work.

## ETHICS STATEMENT

This work adheres to the ICLR Code of Ethics. All datasets used in our experiments are publicly available and were collected by third parties in compliance with applicable laws and ethical guidelines. No human subjects, personally identifiable information, or sensitive data were involved. Our research does not contain or promote harmful content, and all methodologies are intended for benign, academic, and scientific purposes.

## REPRODUCIBILITY STATEMENT

We have made every effort to ensure the reproducibility of our results. All model architectures, training procedures, hyperparameters, and evaluation metrics are described in detail in Sections 4 of the main paper and in the Appendix. The source code, along with configuration files and instructions for reproducing the experiments, will be released soon.

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

## A APPENDICES

### A.1 PROOF OF PROPOSITION 3.2

Recalling the weight, distortion and bias definitions in equations equation 5 and equation 6, respectively, we have:

$$
\begin{aligned}
\mathbb{E}\left[R^q(f)\right] &= \mathbb{E}_{\mathcal{X}_m \sim \tilde{\mathbb{Q}}_t}\left[\frac{1}{m}\sum_{x \in \mathcal{X}_m}\beta(x)\ell\left(y^t(x), f(x)\right)\right] \\
&= \mathbb{E}_{x \sim \mathbb{X}}\left[\mathbb{E}_{y^t(x)|x}[\ell(y^t(x), f(x))]\right] \\
&= \mathbb{E}_{x \sim \mathbb{X}}\left[p(x)\ell\left(f_{\text{false}}(x), f(x)\right) + (1-p(x))\ell\left(y, f(x)\right)\right] \\
&= \mathbb{E}_{x \sim \mathbb{X}}\left[\ell\left(y, f(x)\right)\right] + \mathbb{E}_{x \sim \mathbb{X}}\left[p(x) \cdot \left(\ell\left(f_{\text{false}}(x), f(x)\right) - \ell\left(y, f(x)\right)\right)\right] \\
&= \mathbb{E}_{x \sim \mathbb{X}}\left[\ell\left(y, f(x)\right)\right] + \mathbb{E}_{x \sim \mathbb{X}}\left[p(x) \cdot \left(\frac{\ell\left(f_{\text{false}}(x), f(x)\right)}{\ell\left(y, f(x)\right)} - 1\right) \cdot \ell\left(y, f(x)\right)\right] \\
&= \mathbb{E}_{x \sim \mathbb{X}}\left[\ell\left(y, f(x)\right)\right] + \mathbb{E}_{x \sim \mathbb{X}}\left[p(x) \cdot (d_f(x) - 1) \cdot \ell\left(y, f(x)\right)\right] \\
&= R(f) + \text{Bias}(f).
\end{aligned}
\tag{9}
$$

### A.2 PROOF OF PROPOSITION 3.3

We have

$$
\begin{aligned}
\mathbb{E}\left[R^w(f)\right] &= \mathbb{E}_{\mathcal{X}_m \sim \tilde{\mathbb{Q}}_t}\left[\frac{1}{m}\sum_{x \in \mathcal{X}_m}w_f(x)\,\ell\left(y^t(x), f(x)\right)\right] \\
&= \mathbb{E}_{x \sim \mathbb{X}}\left[\mathbb{E}_{y^t(x)|x}\left[w_f(x)\ell(y^t(x), f(x))\right]\right] \\
&= \mathbb{E}_{x \sim \mathbb{X}}\left[w_f(x) \cdot (p(x)\ell\left(f_{\text{false}}(x), f(x)\right) + (1-p(x))\ell\left(y, f(x)\right))\right] \\
&= \mathbb{E}_{x \sim \mathbb{X}}\left[\frac{p(x)\ell\left(f_{\text{false}}(x), f(x)\right) + (1-p(x))\ell\left(y, f(x)\right)}{1 + p(x) \cdot (d_f(x) - 1)}\right] \\
&= \mathbb{E}_{x \sim \mathbb{X}}\left[\frac{\ell\left(y, f(x)\right) + \ell\left(y, f(x)\right) \cdot p(x) \cdot (d_f(x) - 1)}{1 + p(x) \cdot (d_f(x) - 1)}\right]. \\
&= R(f).
\end{aligned}
\tag{10}
$$

### A.3 STUDYING THE MSE OF A FIXED PREDICTION

In this section, we study the Mean-Squared-Error (MSE) of a fixed prediction $f$ for an arbitrary object $x \in \mathcal{X}$

In what follows. we theoretically demonstrate the case when the weighting scheme could potentially underperform the standard unweighted approach from a bias-variance perspective. The derivations also provide theoretical justification for projecting the weights to the $[0, 1]$ interval (recall Line 16 of Algorithm 1). Formally, for the MSE of standard unweighted approach $\text{MSE}(x)$ and the MSE of our proposed SAW scheme $\text{MSE}^w(x)$, the following equations hold

$$
\begin{aligned}
\text{MSE}(x) &= \mathbb{E}_{y^t(x)|x}\left[\left(\ell\left(y, f(x)\right) - \ell(y^t(x), f(x))\right)^2\right], \\
\text{MSE}^w(x) &= \mathbb{E}_{y^t(x)|x}\left[\left(\ell\left(y, f(x)\right) - w_f(x)\ell(y^t(x), f(x))\right)^2\right].
\end{aligned}
\tag{11}
$$

Recalling the definition of $d_f(x)$ in equation 5, we have the following proposition.

**Proposition A.1.** *Let* $\ell : \mathbb{R}^C \times \mathbb{R}^C \to \mathbb{R}_+$ *be a bounded loss function. Fix* $x \in \mathcal{X}$ *and a predictor* $f : \mathcal{X} \to \mathbb{R}^C$. *We have* $\text{MSE}(x) < \text{MSE}^w(x)$ *if and only if:*

*1.* $d_f(x) < 1/2$;

*2.* $p(x) \in \left(0, \frac{1 - 2 \cdot d_f(x)}{(1 - d_f(x))^2}\right)$.

Proof: We have:

$$\text{MSE}(x) = \mathbb{E}_{y^t(x)|x}\left[\left(\ell\left(y, f(x)\right) - \ell(y^t(x), f(x))\right)^2\right]$$

$$= p(x)\left(\ell\left(y, f(x)\right) - \ell\left(f_{\text{false}}(x), f(x)\right)\right)^2$$

$$= p(x)\ell\left(y, f(x)\right)^2\left(1 - d_f(x)\right)^2$$

and

$$\text{MSE}^w(x) = \mathbb{E}_{y^t(x)|x}\left[\left(\ell\left(y, f(x)\right) - w_f(x)\ell(y^t(x), f(x))\right)^2\right]$$

$$= (1 - p(x))\ell\left(y, f(x)\right)^2\left(1 - w_f(x)\right)^2$$

$$+ p(x)\left(\ell\left(y, f(x)\right) - w_f(x)\ell\left(f_{\text{false}}(x), f(x)\right)\right)^2$$

$$= (1 - p(x))\ell\left(y, f(x)\right)^2\left(1 - w_f(x)\right)^2$$

$$+ p(x)\ell\left(y, f(x)\right)^2\left(1 - w_f(x)d_f(x)\right)^2.$$

Proposition A.1 implies that when it is less likely for the teacher to corrupt the label of data sample $x$, i.e., $p(x)$ is sufficiently small, and the prediction of the student is close enough to the corrupted label (i.e., when $d_f(x)$ is small enough), we can use the unweighted estimator instead of the weighted one from a bias-variance trade-off perspective, as the former has smaller MSE in this case. This observation aligns well with our method as we always project $w_f(x)$ to $[0, 1]$ ($w_f(x) > 1$ iff $d_f(x) < 1$ and $p(x) > 0$).

## A.4 THE ALGORITHMS

---

**Algorithm 1** Weight Estimation

---

**Input:** Samples in $\mathcal{X}_m$ and validation dataset size $\mathcal{S}_v$
**Output:** Weight $w(x)$ for $x$ in $\mathcal{X}_m$
$U \leftarrow \emptyset, k \leftarrow \lceil \sqrt{v}/2 \rceil$
**for** $(x, y) \in \mathcal{S}_v$ **do**
    $X \leftarrow (\text{Confidence}(y^t(x)), \text{Confidence}(f(x)))$
    **if** $y^t(x) = y$ then **then**
        $(p, d) \leftarrow (0, 1)$
    **else**
        $(p, d) \leftarrow \left(1, \frac{\ell(y^t(x), f(x))}{\ell(y, f(x))}\right)$
    **end if**
    $Y \leftarrow (p, d)$
    $U \leftarrow U \cup \{(X, Y)\}$
**end for**
Weights $= \emptyset$
**for** every $x \in \mathcal{X}_m$ **do**
    Query $\leftarrow (\text{Confidence}(y^t(x)), \text{Confidence}(f(x)))$
    $(\hat{p}, \hat{d}) \leftarrow k\text{-NN}(U, \text{Query})$
    $w(x) \leftarrow \min\left\{1, \frac{\beta(x)}{1 + \hat{p}(\hat{d} - 1)}\right\}$
**end for**

---

**Algorithm 2** The SAW Framework

---

**Input:** Training dataset $\mathcal{X}$ and validation dataset $\mathcal{S}_v$
**Output:** Sampled dataset $\mathcal{X}_m$ and $w(x)$ for $x$ in $\mathcal{X}_m$.
/* **Data Sampling** */
Select teacher samples with the highest confidence for each sequence
Calculate entropy difference $h(x)$ for each sample in $\mathcal{X}$
**for** every $x \in \mathcal{X}$ **do**
    $q(x) \leftarrow \frac{h(x) + \epsilon}{\sum_{x' \in \mathcal{X}} (h(x') + \epsilon)}$
**end for**
Obtain $\mathcal{X}_m$ by sampling $m$ data from $\mathcal{X}$ according to distribution $q(x)$
/* **Weight Estimation** */
Estimate $w(x)$ according to Algorithm 1

---

### A.5 DISCUSSIONS ON ESTIMATING (6)

We provide some discussions on the estimate of (6) and Algorithm 1 for practical implementations.

First, the number $k$ of neighbors for weight estimation is set to $k = \frac{\sqrt{v}}{2}$ in Algorithm 1, where $v$ is the size of the validation dataset. This is because choosing $k = \alpha \left( v^{2/(2+\dim)} \right)$, where $v$ is the size of the validation dataset and $\dim$ is the dimension of the underlying metric space ($\dim = 2$ in our case), is asymptotically optimal, and $\alpha = 1/2$ is a popular choice for the constant used in practice Cover & Hart (1967), Hastie (2009).

Second, notice that equation 6 indicates that the weight of an example can exceed 1 if and only if the corresponding distortion value in equation 6 for that example is less than 1. This situation might arise, for instance, when both the student and teacher have the same (or very similar) inaccurate predictions for a given example. In such cases, the weight value in equation 6 suggests that the risk associated with this example should be higher than the low value implied by the unweighted loss function. However, since ground-truth labels are unavailable during training and only the teacher's inaccurate prediction is accessible, assigning a weight greater than 1 in this scenario could lead to the model overfitting the inaccurate label. To prevent this issue, we project all weights onto the $[0, 1]$ interval. Additionally, Appendix A.3 provides further justification for this projection by analyzing the benefits of constraining weights for low-distortion examples within $[0, 1]$ range using MSE considerations.

At last, according to equation 6, the weight of an example is a function of the predictor, which, in the case of neural networks, corresponds to the model parameters. This means that ideally we should update our weights assignment every time the parameters get updated during training. However, we empirically observe that even estimating the weight assignment only once during the whole training process is sufficient to achieve our objectives. As a result, our method adds minimal overhead to the standard training process.

### A.6 ABLATION STUDY RESULTS OF $k$ ON THE NUSCENES DATASET.

Table 3: Ablation study results of $k$ using the proposed SAW (ent) framework on the nuScenes dataset.

| Pre-trained | ResNet-50 | | |
| | 75% | 70% | 65% |
| --- | --- | --- | --- |
| $k = \sqrt{v}/4$ | 81.32 ($\pm$ 0.14) | 77.99 ($\pm$ 0.16) | 69.78 ($\pm$ 0.20) |
| $k = \sqrt{v}/2$ | 81.37 ($\pm$ 0.15) | 78.07 ($\pm$ 0.17) | 69.87 ($\pm$ 0.23) |
| $k = \sqrt{v}$ | 81.34 ($\pm$ 0.14) | 78.02 ($\pm$ 0.17) | 69.82 ($\pm$ 0.24) |

## A.7 ADDITIONAL EXPERIMENTAL RESULTS ON THE NUMBER OF SELECTED SAMPLES ON THE NUSCENES DATASET

Table 4: Performance in the Boston Seaport scenario on the nuScenes dataset. Test accuracies under different numbers of selected training samples $m$ after 100 epochs. The pre-trained model accuracy for the Boston scenario is 75%. The value in the bracket represents the standard deviation. We use boldface to highlight the highest accuracy improvement for each pre-trained model, except for the fully-supervised learning benchmark.

| Num. Samples | ResNet-50 | | | Swin-T | | |
|---|---|---|---|---|---|---|
| | $m = 32000$ | $m = 24000$ | $m = 16000$ | $m = 32000$ | $m = 24000$ | $m = 16000$ |
| w/o SAW | 76.18↑(± 0.28) | 75.36↑(± 0.15) | 75.31↑(± 0.18) | 76.69↑(± 0.26) | 76.06↑(± 0.25) | 75.87↑(± 0.28) |
| AdaCon | 77.88↑(± 0.19) | 76.93↑(± 0.23) | 76.27↑(± 0.28) | 76.85↑(± 0.33) | 76.24↑(± 0.35) | 75.73↑(± 0.37) |
| PLUE | 77.94↑(± 0.12) | 76.48↑(± 0.23) | 75.84↑(± 0.24) | 77.43↑(± 0.24) | 76.95↑(± 0.26) | 76.51↑(± 0.25) |
| RoTTA | 78.67↑(± 0.21) | 77.93↑(± 0.30) | 77.22↑(± 0.29) | — | — | — |
| C-SFDA | 78.71↑(± 0.22) | 77.54↑(± 0.27) | 76.63↑(± 0.30) | 77.75↑(± 0.18) | 77.36↑(± 0.20) | 76.97↑(± 0.22) |
| UPA | 76.87↑(± 0.36) | 76.03↑(± 0.35) | 75.47↑(± 0.38) | 78.24↑(± 0.26) | 77.86↑(± 0.27) | 77.35↑(± 0.26) |
| AnCon | 79.23↑(± 0.29) | 78.71↑(± 0.33) | 78.16↑(± 0.32) | 80.38↑(± 0.27) | 80.02↑(± 0.25) | 79.66↑(± 0.28) |
| W-only (ent) | 77.22↑(± 0.22) | 76.79↑(± 0.13) | 76.54↑(± 0.18) | 77.82↑(± 0.23) | 76.90↑(± 0.17) | 76.45↑(± 0.23) |
| S-only (ent) | 80.76↑(± 0.23) | 80.42↑(± 0.17) | 79.95↑(± 0.25) | 89.48↑(± 0.11) | 89.16↑(± 0.16) | 88.87↑(± 0.15) |
| **SAW** (ent) | **81.37↑(± 0.15)** | **80.91↑(± 0.16)** | 80.45↑(± 0.25) | **90.25↑(± 0.15)** | **89.79↑(± 0.09)** | 89.57↑(± 0.24) |
| W-only (ms) | 77.23↑(± 0.11) | 76.66↑(± 0.18) | 76.41↑(± 0.20) | 77.80↑(± 0.18) | 76.97↑(± 0.22) | 76.55↑(± 0.19) |
| S-only (ms) | 80.69↑(± 0.24) | 80.07↑(± 0.21) | 79.95↑(± 0.19) | 89.50↑(± 0.22) | 89.29↑(± 0.24) | 88.78↑(± 0.22) |
| **SAW** (ms) | 81.21↑(± 0.25) | 80.84↑(± 0.21) | **80.49↑(± 0.23)** | 90.22↑(± 0.16) | 89.83↑(± 0.15) | **89.57↑(± 0.16)** |
| FSL | 88.66↑(± 0.12) | 87.56↑(± 0.19) | 86.83↑(± 0.28) | 92.30↑(± 0.06) | 91.85↑(± 0.16) | 91.36↑(± 0.24) |

## A.8 Performance on KITTI Dataset

We also include one more dataset KITTI, a famous dataset for autonomous driving, for performance evaluation. We evaluate the performance by varying the pre-trained model accuracy and the number of training samples $m$, which is the same as in the nuScenes dataset.

The simulation results demonstrate consistent and substantial performance improvements of the SAW framework compared to pre-trained models across diverse settings. Furthermore, the results also reveal that our SAW and S-only (not requiring a public dataset) framework outperforms all other TTA baselines. It is worth mentioning that some baselines, including W-only and TTA baselines, may result in even worse performance compared with pre-trained model accuracy. However, our SAW and S-only can always improve model performance after adaptation.

Table 5: Performance on the KITTI dataset. Test accuracies after 100 epochs under varied pre-trained model accuracy (i.e., 75%, 70%, and 65%). The value in the bracket represents the standard deviation. We use bold symbols to highlight the highest accuracy improvement for each pre-trained model, except for the fully-supervised fine-tuning benchmark. ↑ and ↓ indicate performance improvement/degradation compared with the pre-trained model.

| Pre-train | ResNet-50 | | | Swin-T | | |
|---|---|---|---|---|---|---|
| | 75% | 70% | 65% | 75% | 70% | 65% |
| w/o SAW | 76.20↑(± 0.24) | 72.14↑(± 0.28) | 60.46↓(± 0.43) | 75.98↑(± 0.31) | 72.24↑(± 0.28) | 41.42↓(± 0.38) |
| AdaCon | 77.03↑(± 0.30) | 73.38↑(± 0.25) | 60.74↓(± 0.32) | 76.32↑(± 0.28) | 73.51↑(± 0.30) | 59.78↓(± 0.43) |
| PLUE | 77.56↑(± 0.26) | 73.05↑(± 0.31) | 67.32↑(± 0.35) | 76.88↑(± 0.24) | 73.40↑(± 0.27) | 65.32↑(± 0.34) |
| RoTTA | 78.16↑(± 0.24) | 74.51↑(± 0.28) | 67.75↑(± 0.39) | — | — | — |
| C-SFDA | 78.00↑(± 0.20) | 74.98↑(± 0.32) | 67.79↑(± 0.25) | 77.15↑(± 0.23) | 72.87↑(± 0.24) | 67.64↑(± 0.29) |
| UPA | 76.39↑(± 0.37) | 73.44↑(± 0.27) | 59.77↓(± 0.53) | 76.88↑(± 0.19) | 72.57↑(± 0.23) | 67.19↑(± 0.28) |
| AnCon | 78.46↑(± 0.25) | 75.27↑(± 0.45) | 68.62↑(± 0.37) | 78.74↑(± 0.24) | 74.11↑(± 0.27) | 69.04↑(± 0.31) |
| W-only (ent) | 77.29↑(± 0.11) | 73.38↑(± 0.20) | 66.07↑(± 0.28) | 77.76↑(± 0.14) | 75.32↑(± 0.22) | 47.32↓(± 0.23) |
| S-only (ent) | 80.97↑(± 0.23) | 77.84↑(± 0.29) | 68.73↑(± 0.33) | 84.22↑(± 0.13) | 83.38↑(± 0.17) | 83.09↑(± 0.24) |
| **SAW** (etr) | **81.41**↑(**± 0.14**) | 78.22↑(± 0.16) | 69.11↑(± 0.24) | **84.88**↑(**± 0.12**) | **83.95**↑(**± 0.08**) | 83.48↑(± 0.15) |
| W-only (ms) | 77.18↑(± 0.13) | 73.42↑(± 0.15) | 65.92↑(± 0.23) | 77.19↑(± 0.22) | 75.16↑(± 0.20) | 48.01↓(± 0.29) |
| S-only (ms) | 80.88↑(± 0.22) | 78.02↑(± 0.26) | 68.98↑(± 0.23) | 84.15↑(± 0.16) | 83.50↑(± 0.16) | 82.96↑(± 0.25) |
| **SAW** (ms) | 81.33↑(± 0.12) | **78.46**↑(**± 0.17**) | **69.41**↑(**± 0.19**) | 84.72↑(± 0.18) | 83.84↑(± 0.23) | **83.56**↑(**± 0.17**) |
| FSL | 86.98↑(± 0.15) | 86.12↑(± 0.27) | 85.25↑(± 0.19) | 88.54↑(± 0.12) | 87.69↑(± 0.21) | 86.55↑(± 0.33) |

Table 6: Performance on the KITTI dataset. Test accuracies under different numbers of selected training samples $m$ after 100 epochs. The pre-trained model accuracy on KITTI is 75%. The value in the bracket represents the standard deviation. We use boldface to highlight the highest accuracy improvement for each pre-trained model, except for the fully-supervised learning benchmark.

| Num. Samples | ResNet-50 | | | Swin-T | | |
|---|---|---|---|---|---|---|
| | $m = 32000$ | $m = 24000$ | $m = 16000$ | $m = 32000$ | $m = 24000$ | $m = 16000$ |
| w/o SAW | 76.20↑(± 0.24) | 75.82↑(± 0.34) | 75.37↑(± 0.31) | 75.98↑(± 0.31) | 74.71↑(± 0.23) | 74.24↓(± 0.24) |
| AdaCon | 77.03↑(± 0.30) | 76.75↑(± 0.21) | 75.89↑(± 0.24) | 76.32↑(± 0.28) | 75.94↑(± 0.26) | 75.23↑(± 0.34) |
| PLUE | 77.56↑(± 0.19) | 77.11↑(± 0.25) | 76.74↑(± 0.20) | 76.88↑(± 0.24) | 76.51↑(± 0.22) | 75.86↑(± 0.28) |
| RoTTA | 78.16↑(± 0.24) | 77.72↑(± 0.26) | 77.30↑(± 0.23) | — | — | — |
| C-SFDA | 78.00↑(± 0.20) | 77.53↑(± 0.22) | 77.13↑(± 0.27) | 77.15↑(± 0.23) | 76.73↑(± 0.25) | 76.20↑(± 0.29) |
| UPA | 76.39↑(± 0.37) | 76.01↑(± 0.35) | 75.55↑(± 0.38) | 76.88↑(± 0.19) | 76.30↑(± 0.21) | 75.85↑(± 0.27) |
| AnCon | 78.46↑(± 0.25) | 78.05↑(± 0.23) | 77.68↑(± 0.26) | 78.74↑(± 0.24) | 78.23↑(± 0.19) | 77.88↑(± 0.21) |
| W-only (ent) | 77.29↑(± 0.11) | 76.84↑(± 0.23) | 76.06↑(± 0.34) | 77.76↑(± 0.14) | 76.97↑(± 0.23) | 76.43↑(± 0.26) |
| S-only (ent) | 80.97↑(± 0.23) | 80.76↑(± 0.17) | 80.05↑(± 0.33) | 84.22↑(± 0.13) | 83.95↑(± 0.11) | 83.47↑(± 0.19) |
| **SAW** (ent) | **81.41**↑(**± 0.14**) | 81.02↑(± 0.21) | **80.68**↑(**± 0.24**) | **84.88**↑(**± 0.12**) | 84.47↑(± 0.15) | **83.98**↑(**± 0.23**) |
| W-only (ms) | 77.18↑(± 0.13) | 76.78↑(± 0.10) | 76.32↑(± 0.24) | 77.19↑(± 0.22) | 76.68↑(± 0.18) | 76.18↑(± 0.33) |
| S-only (ms) | 80.88↑(± 0.22) | 80.44↑(± 0.17) | 79.93↑(± 0.23) | 84.15↑(± 0.16) | 83.73↑(± 0.22) | 83.39↑(± 0.31) |
| SAW (ms) | 81.33↑(± 0.12) | **81.10**↑(**± 0.15**) | 80.31↑(± 0.18) | 84.72↑(± 0.18) | **84.25**↑(**± 0.25**) | 83.84↑(± 0.15) |
| FSL | 86.98↑(± 0.15) | 86.10↑(± 0.24) | 85.76↑(± 0.27) | 88.54↑(± 0.12) | 87.76↑(± 0.18) | 87.28↑(± 0.29) |

## A.9 PERFORMANCE ON THE EMBODIED AGENT SCENARIO

To further demonstrate the broad applicability of our methodology, we add experiments in the embodied agent domain by still adapting the pre-trained models from the Singapore Onenorth scenario in the nuScenes dataset. We select frames from the widely investigated tracking datasets, i.e., JRDB Martin-Martin et al. (2021) (collected by robots), MOT Voigtlaender et al. (2019), and LaSOT Fan et al. (2019) datasets, with classes, including car, pedestrian, robot, drone, basketball, bottle, dog, and bicycle, to form a dataset for robotic scenarios. We selected 800 sequences and extracted over 40k images used for model adaptation and 5k images for testing.

It can be seen that our proposed method consistently outperforms the non-adaptive model and the other baselines. These new experiments in an entirely new domain have demonstrated the broad applicability of our method.

Table 7: Performance on the embodied agent scenario. Test accuracies after 100 epochs under varied pre-trained model accuracy for this scenario (i.e., 75%, 70%, and 65%). The value in the bracket represents the standard deviation. We use bold symbols to highlight the highest accuracy improvement for each pre-trained model, except for the fully-supervised learning benchmark.

| | ResNet-50 | | | Swin-T | | |
|---|---|---|---|---|---|---|
| Pre-train | 75% | 70% | 65% | 75% | 70% | 65% |
| w/o SAW | 76.03↑(± 0.28) | 71.06↑(± 0.29) | 60.07↓(± 0.34) | 76.03↑(± 0.28) | 71.32↑(± 0.31) | 38.47↓(± 0.35) |
| AdaCon | 76.85↑(± 0.26) | 72.63↑(± 0.25) | 65.63↑(± 0.29) | 76.81↑(± 0.22) | 72.04↑(± 0.24) | 58.48↑(± 0.27) |
| PLUE | 77.01↑(± 0.25) | 72.88↑(± 0.27) | 65.75↑(± 0.33) | 76.83↑(± 0.19) | 72.79↑(± 0.21) | 60.27↑(± 0.30) |
| RoTTA | 77.76↑(± 0.18) | 74.00↑(± 0.20) | 67.36↑(± 0.23) | — | — | — |
| C-SFDA | 77.93↑(± 0.22) | 73.66↑(± 0.22) | 68.53↑(± 0.31) | 77.12↑(± 0.17) | 73.02↑(± 0.26) | 66.88↑(± 0.28) |
| UPA | 77.57↑(± 0.27) | 72.61↑(± 0.32) | 62.23↓(± 0.31) | 76.62↑(± 0.30) | 72.70↑(± 0.29) | 66.51↑(± 0.33) |
| Ancon | 78.33↑(± 0.21) | 74.01↑(± 0.27) | 67.43↑(± 0.30) | 78.89↑(± 0.16) | 75.05↑(± 0.26) | 68.84.89↑(± 0.25) |
| W-only (ent) | 76.80↑(± 0.16) | 72.57↑(± 0.23) | 67.34↑(± 0.25) | 77.38↑(± 0.21) | 72.06↑(± 0.26) | 39.09↓(± 0.35) |
| S-only (ent) | 79.65↑(± 0.17) | 77.36↑(± 0.19) | 68.98↑(± 0.26) | 86.94↑(± 0.21) | 86.67↑(± 0.23) | 86.65↑(± 0.24) |
| **SAW** (ent) | **80.21**↑(**± 0.16**) | 77.74↑(± 0.21) | **69.45**↑(**± 0.23**) | **87.41**↑(**± 0.14**) | 87.29↑(± 0.16) | 87.18↑(± 0.19) |
| W-only (ms) | 76.78↑(± 0.21) | 72.50↑(± 0.24) | 67.39↑(± 0.27) | 77.31↑(± 0.12) | 71.98↑(± 0.22) | 39.17↓(± 0.18) |
| S-only (ms) | 79.59↑(± 0.15) | 77.28↑(± 0.21) | 68.92↑(± 0.19) | 86.82↑(± 0.13) | 86.60↑(± 0.16) | 86.46↑(± 0.25) |
| **SAW** (ms) | 80.06↑(± 0.14) | **77.82**↑(**± 0.18**) | 69.41↑(± 0.21) | 87.38↑(± 0.23) | **87.35**↑(**± 0.19**) | **87.23**↑(**± 0.21**) |
| FSL | 86.14↑(± 0.15) | 85.77↑(± 0.16) | 84.94↑(± 0.20) | 89.46↑(± 0.10) | 89.26↑(± 0.17) | 89.21↑(± 0.20) |

## A.10 Performance on the sequence construction

We illustrate the impact of different sequence construction methods in Table 8. The "Well Constructed" approach refers to training on perfectly constructed sequences, based on ground-truth object IDs from the dataset, whereas the "Based on Tracking" approach constructs sequences using the object tracking algorithm (the Average Multi-object Tracking Accuracy (AMOTA) on the test dataset is 67.3%) Qin et al. (2024). Results indicate that off-the-shelf tracking algorithms are sufficient for sequence construction, with only negligible degradation in performance compared to the ideal, ground-truth-based object associations.

Table 8: Performance in the Boston Seaport scenario on the nuScenes dataset with well-constructed sequence and sequences constructed by off-the-shelf object tracking algorithm.

| Sequence Construction | Pre-trained | ResNet-50 | | |
| --- | --- | --- | --- | --- |
| | | 75% | 70% | 65% |
| Well Constructed | w/o SAW | 76.18↑(± 0.28) | 72.05↑(± 0.28) | 62.17↓(± 0.37) |
| | AdaCon | 77.88↑(± 0.19) | 73.63↑(± 0.15) | 66.51↑(± 0.18) |
| | PLUE | 77.94↑(± 0.12) | 73.82↑(± 0.21) | 67.28↑(± 0.29) |
| | RoTTA | 78.67↑(± 0.21) | 75.13↑(± 0.33) | 68.20↑(± 0.31) |
| | C-SFDA | 78.71↑(± 0.22) | 75.37↑(± 0.26) | 68.34↑(± 0.34) |
| | UPA | 76.87↑(± 0.36) | 74.13↑(± 0.34) | 57.60↓(± 0.48) |
| | AnCon | 79.23↑(± 0.29) | 75.61↑(± 0.36) | 68.23↑(± 0.38) |
| | W-only (ent) | 77.22↑(± 0.22) | 73.63↑(± 0.21) | 65.43↑(± 0.17) |
| | S-only (ent) | 80.76↑(± 0.23) | 76.78↑(± 0.26) | 69.16↑(± 0.21) |
| | **SAW** (ent) | **81.37**↑(± **0.15**) | **78.07**↑(± **0.17**) | 69.87↑(± 0.23) |
| | W-only (ms) | 77.23↑(± 0.11) | 73.41↑(± 0.34) | 65.36↑(± 0.25) |
| | S-only (ms) | 80.69↑(± 0.24) | 76.68↑(± 0.30) | 69.48↑(± 0.26) |
| | **SAW** (ms) | 81.21↑(± 0.25) | 77.91↑(± 0.16) | **70.02**↑(± **0.18**) |
| | FSL | 88.66↑(± 0.12) | 87.83↑(± 0.18) | 86.87↑(± 0.25) |
| Based on Tracking | w/o SAW | 75.53↑(± 0.24) | 71.41↑(± 0.26) | 61.48↑(± 0.39) |
| | AdaCon | 76.45↑(± 0.14) | 72.72↑(± 0.18) | 65.46↑(± 0.19) |
| | PLUE | 77.16↑(± 0.16) | 73.04↑(± 0.25) | 66.80↑(± 0.26) |
| | RoTTA | 77.35↑(± 0.24) | 73.85↑(± 0.23) | 67.09↑(± 0.27) |
| | C-SFDA | 78.03↑(± 0.21) | 73.99↑(± 0.28) | 67.60↑(± 0.31) |
| | UPA | 76.15↑(± 0.30) | 73.31↑(± 0.31) | 56.81↑(± 0.39) |
| | AnCon | 78.88↑(± 0.24) | 75.04↑(± 0.33) | 67.93↑(± 0.32) |
| | W-only (ent) | 76.95↑(± 0.20) | 73.42↑(± 0.22) | 65.11↑(± 0.23) |
| | S-only (ent) | 80.43↑(± 0.16) | 76.46↑(± 0.20) | 68.92↑(± 0.23) |
| | **SAW** (ent) | **81.06**↑(± **0.14**) | 77.69↑(± 0.18) | 69.55↑(± 0.25) |
| | W-only (ms) | 76.82↑(± 0.15) | 73.03↑(± 0.27) | 65.03↑(± 0.29) |
| | S-only (ms) | 80.36↑(± 0.18) | 76.29↑(± 0.24) | 69.13↑(± 0.30) |
| | **SAW** (ms) | 80.92↑(± 0.16) | **77.73**↑(± **0.18**) | **69.77**↑(± **0.21**) |
| | FSL | 88.34↑(± 0.13) | 87.40↑(± 0.20) | 86.51↑(± 0.28) |

Table 9: Performance in the Boston Seaport scenario on the nuScenes dataset with well-constructed sequences and sequences constructed by an off-the-shelf object tracking algorithm.

| Sequence Construction | Pre-trained | Swin-T | | |
|---|---|---|---|---|
| | | 75% | 70% | 65% |
| Well Constructed | w/o SAW | 76.69↑(± 0.26) | 72.55↑(± 0.25) | 40.23↓(± 0.43) |
| | AdaCon | 76.85↑(± 0.33) | 72.83↑(± 0.47) | 62.28↓(± 0.61) |
| | PLUE | 77.43↑(± 0.24) | 73.55↑(± 0.30) | 59.91↓(± 0.35) |
| | C-SFDA | 77.75↑(± 0.18) | 72.20↑(± 0.19) | 66.43↑(± 0.25) |
| | UPA | 78.24↑(± 0.26) | 73.15↑(± 0.23) | 68.12↑(± 0.28) |
| | AnCon | 80.38↑(± 0.27) | 76.28↑(± 0.30) | 70.59↑(± 0.32) |
| | W-only (ent) | 77.82↑(± 0.23) | 74.83↑(± 0.12) | 43.96↓(± 0.35) |
| | S-only (ent) | 89.48↑(± 0.11) | 89.23↑(± 0.13) | 89.25↑(± 0.12) |
| | **SAW** (ent) | **90.25**↑(± **0.15**) | 90.18↑(± 0.08) | 90.09↑(± 0.07) |
| | W-only (ms) | 77.80↑(± 0.18) | 74.85↑(± 0.15) | 43.89↓(± 0.30) |
| | S-only (ms) | 89.50↑(± 0.22) | 89.10↑(± 0.32) | 89.01↑(± 0.41) |
| | **SAW** (ms) | 90.22↑(± 0.16) | **90.21**↑(± **0.08**) | **90.04**↑(± **0.23**) |
| | FSL | 92.30↑(± 0.06) | 92.12↑(± 0.15) | 91.98↑(± 0.28) |
| Based on Tracking | w/o SAW | 76.10↑(± 0.25) | 71.96↑(± 0.27) | 39.51↑(± 0.39) |
| | AdaCon | 76.23↑(± 0.26) | 72.16↑(± 0.22) | 60.39↑(± 0.54) |
| | PLUE | 76.88↑(± 0.25) | 73.00↑(± 0.32) | 57.58 ↓(± 0.34) |
| | C-SFDA | 77.11↑(± 0.20) | 71.59↑(± 0.23) | 65.42↑(± 0.27) |
| | UPA | 77.44↑(± 0.28) | 72.61↑(± 0.27) | 67.55↑(± 0.23) |
| | AnCon | 79.73↑(± 0.22) | 75.68↑(± 0.27) | 69.88↑(± 0.35) |
| | W-only (ent) | 77.40↑(± 0.19) | 74.51↑(± 0.16) | 42.66↓(± 0.28) |
| | S-only (ent) | 89.07↑(± 0.14) | 88.66↑(± 0.17) | 88.57↑(± 0.20) |
| | **SAW** (ent) | **89.81**↑(± **0.17**) | 89.67↑(± 0.15) | 89.55↑(± 0.14) |
| | W-only (ms) | 77.26↑(± 0.16) | 74.41↑(± 0.18) | 43.36↓(± 0.27) |
| | S-only (ms) | 89.05↑(± 0.27) | 88.72↑(± 0.31) | 88.68↑(± 0.38) |
| | **SAW** (ms) | 89.88↑(± 0.17) | **89.81**↑(± **0.13**) | **89.73**↑(± **0.24**) |
| | FSL | 91.78↑(± 0.10) | 91.68↑(± 0.16) | 91.55↑(± 0.25) |

### A.11 CLASSIFICATIONS IN OBJECT DETECTION MODELS

Beyond single-label classification, let us consider extending the SAW framework to adapt the classification parts in object detection models. For instance, a well-known object detection model, Faster R-CNN, has a dedicated classification network, which can be fine-tuned while keeping other parts frozen (i.e., freezing the backbone and the regression network). The main difference from our previous design is that the model can classify multiple objects simultaneously in a frame. To employ SAW, we first construct sequences of image frames by their time stamps, where images recorded closely over time are constructed as a sequence. For each object in a sequence, we select the object predictions with the least uncertainty as the teacher output. Then, we measure the entropy difference of all images in the sequence as the *summation* of the entropy difference of each object in this image and the teacher output in the sequence. Essentially, the main distinction from single-label classification lies in the method used to calculate *entropy difference*. The data weighting scheme in (6) can still be applied to make it an unbiased estimator of clean loss. In this way, we can adapt the data sampling and weighting procedure in Section 3.2.1 and 3.2.2 to the classification part of object detection models.

### A.12 EXPERIMENTAL RESULTS FOR FASTER R-CNN

In what follows, we provide experimental results by extending our SAW framework to a well-known object detection model, Faster R-CNN with ResNet-50 as the backbone network. Since the TTA baselines in Section 4 focus on single-label classification, which cannot be easily adapted to object detection models, we only evaluate our approaches, including SAW, S-only, W-only, and w/o SAW, while merely comparing ours with existing TTA schemes in previous single-label classification scenarios.

As shown in Fig. 4, SAW achieves the best test accuracy across varying settings of pre-trained model accuracy and number of data samples. Moreover, SAW and S-only outperform other benchmarks, particularly the pre-trained model, by a substantial margin (e.g., 9%). This suggests that our method can be applied to improve the classification modules of object detection models without manual labeling, thereby directly benefiting a wide range of practical applications with movable agents, including autonomous driving, drones, and robots.

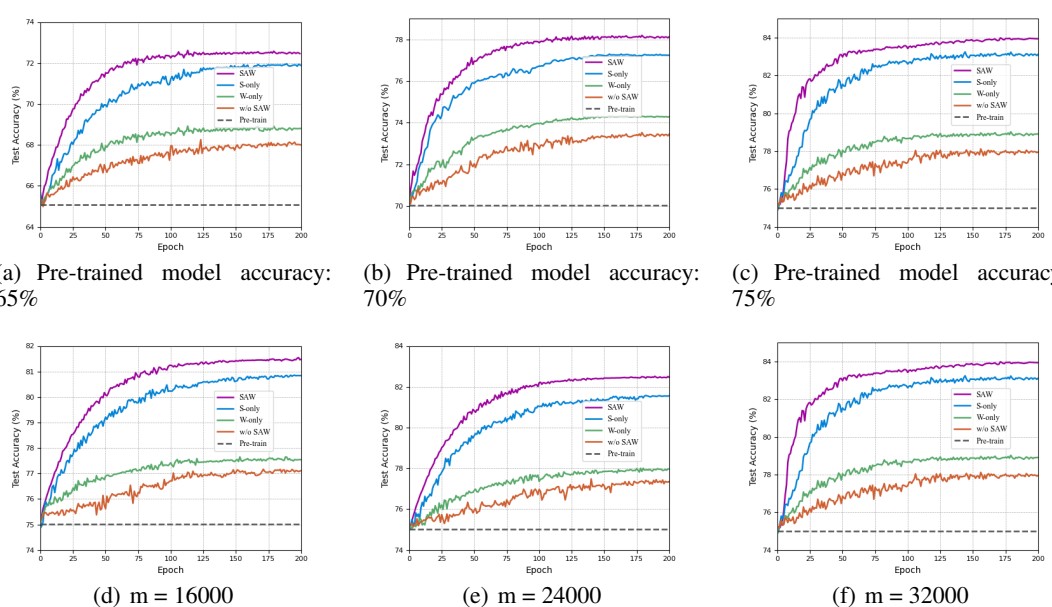

(a) Pre-trained model accuracy: 65%

(b) Pre-trained model accuracy: 70%

(c) Pre-trained model accuracy: 75%

(d) m = 16000

(e) m = 24000

(f) m = 32000

Figure 4: The test accuracy versus epoch with the object detection model Faster R-CNN.

### A.13    ADDITIONAL EXPERIMENTAL RESULTS ON TEST ACCURACY FOR SWIN-T

Considering a Transformer model Swin-T, we visualize the test accuracy in Fig. 5. As can be seen in Fig. 5(a) - Fig. 5(c), the proposed SAW framework consistently outperforms other benchmarks significantly. The key factor behind this performance improvement is our data sampling strategy, which is based on the entropy difference between "teacher" and "student" data samples. This strategy ensures that highly accurate predictions effectively propagate knowledge to less accurate ones, as measured by the entropy difference. In contrast, random sampling approaches, whether they include weighting schemes or not, provide only minor improvements to the pre-trained model. Furthermore, these approaches can lead to overfitting towards corrupted pseudo labels, as evidenced by performance drops as the number of epochs increases. Notably, without appropriate data sampling, performance can deteriorate to levels worse than the pre-trained model's accuracy, as shown in Fig. 5(a). This deterioration often occurs when the pre-trained model's accuracy is low, leading to a higher proportion of corrupted pseudo labels. However, our SAW framework and "sampling only" strategy demonstrate steady performance improvement even in such challenging scenarios.

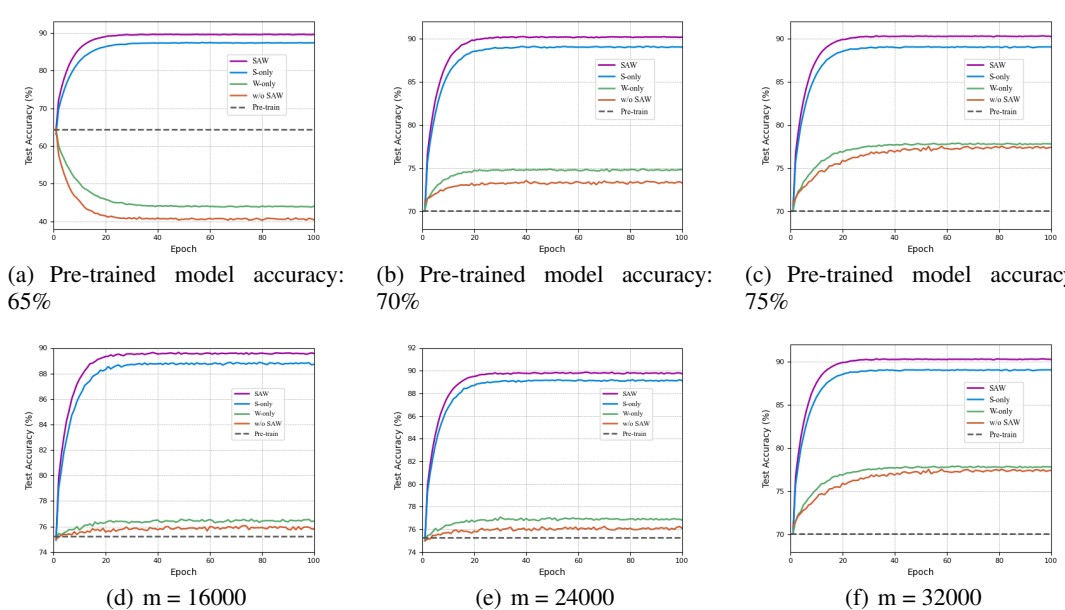

(a) Pre-trained model accuracy: 65%

(b) Pre-trained model accuracy: 70%

(c) Pre-trained model accuracy: 75%

(d) m = 16000

(e) m = 24000

(f) m = 32000

Figure 5: The test accuracy versus epoch with a Transformer Swin-T. Our proposed approach leads to consistently better performance in terms of test-accuracy and convergence speed.

### A.14 ADDITIONAL EXPERIMENTAL RESULTS FOR TEST ACCURACY FOR RESNET-50

We provide more experimental results based on ResNet-50, a well-known convolution neural network, in Fig. 6. As observed, the proposed SAW framework consistently outperforms other benchmarks significantly. Similar to the results for Swin-T, the data sampling strategy plays a very crucial role in enhancing test accuracy, highlighting the importance of sampling data based on entropy differences. Also, the weighting strategy offers consistent improvement over its non-weighted counterpart.

It is important to note that training can sometimes lead to overfitting on corrupted labels. This phenomenon is also observed in the SAW framework, as illustrated in Fig. 6(d) and Fig. 6(e). Fortunately, we observe that 1) our SAW framework substantially mitigates this overfitting compared to other benchmarks, especially the weighted-only or baseline schemes, due to careful data selection and weighting, and 2) the overfitting can be alleviated when the number of training data samples is increased, as evidenced by the absence of performance drop in the SAW framework in Fig. 6(f).

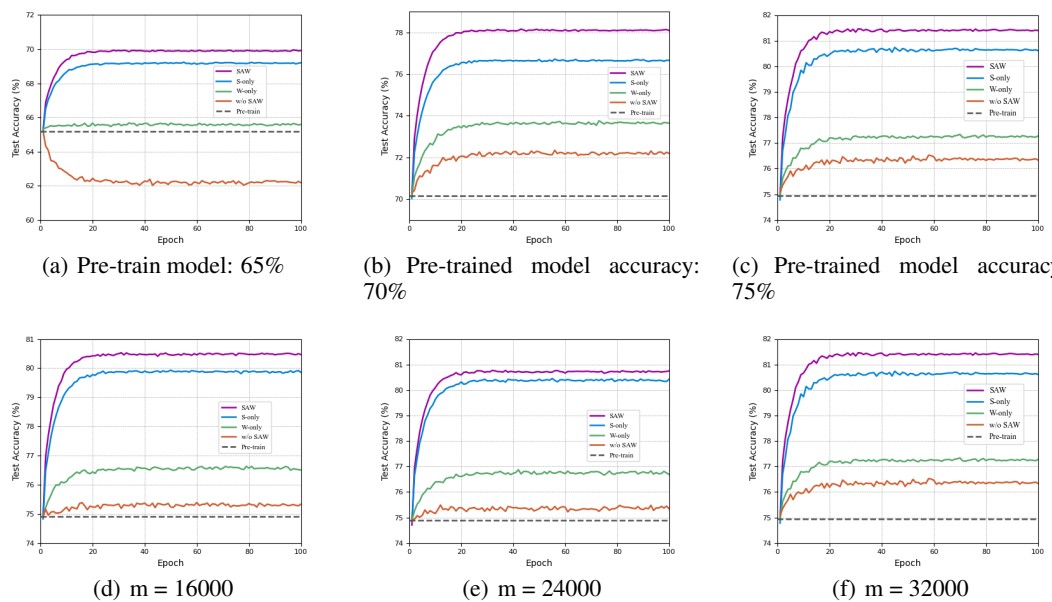

(a) Pre-train model: 65%

(b) Pre-trained model accuracy: 70%

(c) Pre-trained model accuracy: 75%

(d) m = 16000

(e) m = 24000

(f) m = 32000

Figure 6: The test accuracy versus epoch with a ResNet-50 model. Our proposed approach leads to consistently better performance in terms of test-accuracy and convergence speed.

## A.15 THE USE OF LARGE LANGUAGE MODELS

During the preparation of this manuscript, we employed a Large Language Model (ChatGPT, OpenAI) solely as a writing assitance for grammar checking and minor language polishing. All technical content, including research conception, methodology, experiments, data analysis, and conclusions, are entirely the original work of the authors. The LLM did not contribute to research ideation, technical development, or result interpretation. All scientific claims and findings have been independently verified by the authors.

