# OpenReview forum: "Learning by Moving Closer: Adapting Vision Models on Movable Agents without Manual Labeling"
_ICLR.cc/2026/Conference — ICLR 2026 Conference Withdrawn Submission_

### Official Review · Reviewer_XZPq · 2025-10-17

**Soundness:** 3
**Presentation:** 4
**Contribution:** 2
**Rating:** 6
**Confidence:** 3

**Summary:**

This paper proposes a novel test-time adaptation method by leveraging pictures of the same object captured at different positions without manual labeling. It uses the one with the highest confidence as the pseudo ground-truth and proposes the data sampling and re-weighting framework (SAW) to debias the training loss and address the label noise. The experiments show the effectiveness of the method on embodied datasets.

**Strengths:**

1. The paper is well-written and easy to follow with a clear motivation. The experiments and theoretical proofs are comprehensive and sound. The results show clear improvements compared to the existing benchmarks.
2. The idea of using image sequences captured by a moving agent for test-time adaptation is novel.
3. The method is agnostic to model architectures and hyperparameter choices, making it a flexible and general framework.
4. The proposed cross-dataset and cross-embodiment settings are novel and practical for real-world embodied agents.

**Weaknesses:**

1. The authors only validate their approach on improving image classification and the classification head of object detection tasks. Meanwhile, it remains unclear whether such an approach can be easily adapted to more valuable tasks for embodied agents, like 3D detection and reconstruction, and E2E planning, or even VLM finetuning, which have very different training targets and model architectures.
2. The authors only perform evaluation on offline datasets without applying their method to real-world movable agents.
3. Since the authors assume a movable agent in their problem setting, a better solution for perception would be to aggregate information from multiple viewpoints instead of relying on and adapting a single-view model. Therefore, it is questionable whether the method is really useful for movable agents.
4. Even though the S-only variant does not suffer from much performance loss, the assumption of having a validation set for the complete SAW framework is often impractical for movable agents, which often need to be deployed in novel places.

**Questions:**

1. Is it possible to show the application of the method on real-world robots?
2. Does the author have any idea of how to apply the method beyond classification tasks?

I’m not an expert in the domain of test-time adaptation, but I believe it would be very beneficial and I would consider raising my score if the author could provide more results to demonstrate the method's potential on embodied agent tasks.

---

### Official Review · Reviewer_YGAE · 2025-10-29

**Soundness:** 1
**Presentation:** 1
**Contribution:** 1
**Rating:** 2
**Confidence:** 4

**Summary:**

This work proposes a test-time adaptation method for object classification that treats low-variance (high-confidence) predictions as “teachers” to supervise high-variance (low-confidence) ones. While the goal of test-time adaptation for embodied agents is well-motivated, the paper models a video-based object detection problem as an image classification problem, failing to exploit temporal or spatial information. It also assumes that high-confidence predictions are reliable teachers, which risks reinforcing model errors despite the proposed importance-weighting correction. Finally, the notions of “movement” and “learning to see” are used vaguely, leading to a poorly defined contribution.

**Strengths:**

- Strong empirical validation: Experiments with multiple datasets and baselines.

**Weaknesses:**

- Unclear conceptual framing: The notions of “movement” and “learning to see,” are never defined.

- Lack of temporal modeling: The method ignores the temporal dimension of the problem. Although the data come from video sequences, the approach processes each frame independently without modeling motion.

- Lack of novelty: The proposed Sampling-and-Weighting (SAW) framework is a confidence-based pseudo-labeling with importance weighting. Similar ideas have been extensively explored in prior work such as PLUE (Litrico et al., 2023). The propositions in Section 3.2 are derived from standard importance weighting and unbiased risk estimation.

- Weak assumption for pseudo-labeling: The method assumes that low-variance (high-confidence) predictions are good pseudo-labels, which is not theoretically or empirically justified. This assumption may reinforce incorrect predictions despite the proposed importance-weighting correction.

- Weak assumption for viewpoint proximity: The paper assumes that predictions made at closer distances are of higher quality and can therefore serve as reliable teacher signals. However, this may not always hold — if the model was trained primarily on distant views, closer viewpoints could actually degrade performance.

- Errors in in-text citation: Missing parentheses (e.g., line 69), missing spaces between text and citations (line 69), and duplicated references to the same paper (lines 98 and 100).

**Questions:**

- Why do the authors assume that low-variance (high-confidence) predictions are good pseudo-labels?

- Why did the authors choose 65%, 70%, and 75% as the pre-trained accuracy levels in their experiments?

- Given that the authors have access to a sequence of frames for the same object, why is temporal information ignored?

- How do the authors define the notions of “learning to see” and “movement of the agent”?

- Could the authors clarify whether the assumption that closer-distance predictions lead to better results is empirically validated? Wouldd data augmentation during training be sufficient to handle such cases?

---

### Official Review · Reviewer_YH6X · 2025-11-01

**Soundness:** 3
**Presentation:** 3
**Contribution:** 3
**Rating:** 6
**Confidence:** 4

**Summary:**

This paper proposes a test-time adaptation (TTA) framework tailored for movable agents (e.g., robots, autonomous vehicles) that adapt their visual perception models without manual labels. The key idea is that as the agent moves, prediction confidence changes with distance, i.e., closer views tend to yield higher confidence. Authors use these high-confidence predictions as pseudo labels for low-confidence samples. Specifically, they introduce a Sampling-and-Weighting (SAW) framework. 1) Sampling step selects teacher-student pairs based on confidence differences. 2) Weighting step reweights samples using an estimated noise probability and distortion factor, ensuring an unbiased risk estimator even with noisy pseudo labels. Experiments on nuScenes and KITTI datasets show consistent improvements over prior TTA methods for both ResNet-50 and Swin-T. The method also extends to the classification head of Faster R-CNN.

**Strengths:**

- New perspective for TTA: The idea of learning by moving for self-adaptation is intuitive, which naturally fits with embodied AI scenarios.
- The method is insensitive to hyperparameter tuning, contrasting with prior TTA approaches.
- The paper provides clear theoretical justification showing that the weighting step yields an unbiased risk estimator.

**Weaknesses:**

- Narrow task scopes: Experiments are restricted to the classification task (and one detection extension results in appendix). Main novelty would be more convincing with segmentation or 3D perception tasks.
- Dependence on sequence construction: The method assumes reliable sequences associated with objects. Although class-agnostic tracking is discussed, this dependency may limit real-world generalization in terms of available training data.
- Overemphasis on “unbiasedness” claim: While the unbiased loss estimator is theoretically elegant, its practical benefit is not clearly validated through ablation study or visual evidence.

**Questions:**

- How sensitive is the framework to tracking noise or imperfect object association in real deployment (e.g., partial occlusions)?
- How is the validation dataset for weighting obtained in fully unlabeled scenarios? Is public data always available or can it be simulated online?
- Would this approach extend naturally to temporal models (e.g., video transformers or 3D CNNs) without explicit object correspondence?
- What is the computational overhead of computing per-sample weights during adaptation?

---

### Official Review · Reviewer_3kNK · 2025-11-01

**Soundness:** 2
**Presentation:** 3
**Contribution:** 1
**Rating:** 2
**Confidence:** 4

**Summary:**

This paper is about adapting vision models on movable agents by utilizing the natural variation in prediction quality that occurs as the agent changes its viewpoint. The authors introduce a unified data sampling-and-weighting (SAW) framework that uses high-confidence predictions (often from closer views) as pseudo-labels to supervise lower-confidence views, aiming to construct an unbiased estimator of the clean loss.

**Strengths:**

- I like the core intuition of the paper. Exploiting the fact that an agent captures the same object from varying distances, leading to differences in prediction quality, is a compelling approach for self-supervision in the context of robotics and autonomous driving.

- The authors highlight that their approach is model-agnostic and relatively insensitive to hyperparameters, which are advantages over many existing TTA methods.

**Weaknesses:**

My main concerns lie in the novelty of the approach and some contradictions in the problem setting.

- Contradiction in the Unsupervised Setting: The paper positions itself as a method for adaptation "without manual labeling" in new environments. However, the full SAW framework, which provides the theoretical guarantee of an unbiased estimator (Proposition 3.3), relies on a data weighting scheme (Section 3.2.2). Estimating these weights requires calculating the teacher's error probability p(x) and distortion d_f(x). This requirement fundamentally violates the assumptions of unsupervised TTA. If S_v is from the target domain, labels are used. If it's from the source domain, the method is not source-free. If it is an unrelated public dataset (L311) the estimated noise model is unlikely to be accurate for the target domain, weakening the practical relevance of the theoretical claims.

- Incremental Technical Novelty: The technical contribution appears limited. The idea of using confidence for pseudo-labeling is standard in TTA. The main technical component, the weighting scheme to debias the loss, is explicitly adapted from prior work in knowledge distillation (Iliopoulos et al., 2022). The contribution is therefore the application of existing techniques to the "learning by moving" context, rather than a new methodological development.

- The framework critically depends on constructing accurate sequences of the same object across frames. While the authors provide an ablation study suggesting an off-the-shelf tracker suffices, I remain skeptical about real-world robustness. Tracking errors, particularly ID switches in dense environments, are common. Such errors would lead to mismatched teacher-student pairs (e.g., a 'car' supervising a 'truck'), introducing association noise that the framework does not model and which could severely corrupt the adaptation process.

- The method assumes that high confidence correlates with high accuracy. This assumption often fails under distribution shift, which ist the exact scenario TTA addresses. If the model is confidently wrong, even at close range, the framework has no mechanism to reject this incorrect teacher prediction. - potentially leading to confirmation bias.

- The S-only results in Table 1 are very close to the full SAW results. Given that S-only does not require the problematic validation set, what is the justification for the added complexity of the full SAW framework? Does this imply the theoretical benefits of the weighting scheme are negligible in practice?

**Questions:**

- What specific dataset was used for S_v during the cross-region adaptation (Singapore to Boston)? How sensitive is the performance of SAW to the distribution gap between Sv and the target data?

- Why does the W-only scheme perform catastrophically in some settings? (e.g., Table 1, Swin-T 65% accuracy drops to 43.96%) If the weighting provides an unbiased estimator, why does it degrade performance so severely when not combined with the specific sampling strategy?

- The tracker used in A.10 has an AMOTA of 67.3%, implying significant tracking errors. Can you provide a deeper analysis of how specific errors, like ID switches, impact the adaptation performance?

- Have you analyzed the calibration of the pre-trained models in the target domain? I am curious about the actual accuracy of the high-confidence "teacher" predictions. How often is the teacher confidently wrong?

- In Appendix A.3, the weights w(x) are clipped to a maximum of 1. How does this clipping affect the unbiased estimator claim made in Proposition 3.3?

---

### Official Review · Reviewer_35jH · 2025-11-02

**Soundness:** 2
**Presentation:** 2
**Contribution:** 2
**Rating:** 4
**Confidence:** 4

**Summary:**

This paper proposes a new test-time adaptation framework, Learning by Moving Closer (LMC), for adapting pre-trained vision models on movable agents without any manual labeling. The core idea is that when an agent moves, its prediction confidence varies with the viewing distance or angle, for example, closer observations tend to be more accurate. The authors use this property to let high-confidence predictions act as “teachers” to guide low-confidence ones, enabling self-supervised adaptation during movement.

To make this process robust, the paper introduces a Sampling-and-Weighting (SAW) framework. The sampling step selects teacher and student samples based on prediction confidence (e.g., entropy). The weighting step corrects possible teacher errors by assigning importance weights, so that the loss function becomes an unbiased estimator of the clean loss. The method is model-agnostic, hyperparameter-insensitive, and requires no access to source data.

Experiments on nuScenes, KITTI, and embodied-agent datasets show that SAW achieves consistent gains over existing test-time adaptation baselines (e.g., AdaCon, PLUE, RoTTA) on both CNN (ResNet-50) and Transformer (Swin-T) models. The authors also provide theoretical proofs for the unbiased property and demonstrate extensions to object detection models.

**Strengths:**

The paper is well-written and overall well-organized

Authors provided both experimental results and theoretical analysis.

The method is model-agnostic, hyperparameter-insensitive (Section 4.3), and does not require any labeled data or access to the source domain, which is an important property for real-world deployment on robots or vehicles.

**Weaknesses:**

1. The experiments in Table 1 and Table 2 are limited to controlled datasets (nuScenes, KITTI), and the “cross-domain” setting mainly uses embodied-agent data with modest diversity. It remains unclear whether the proposed method would scale or remain stable on large-scale, cluttered, or more dynamic real-world domains.

2. Although the paper mentions extension to object detection (Appendix A.12), the main text focuses only on image classification. The method’s effectiveness on dense prediction tasks (segmentation, depth estimation) is not empirically validated.

3. Section 4.3 explores several factors (e.g., hyperparameters, confidence metrics, sequence construction), but some key sensitivities are missing. For example: How does performance degrade with noisy or inconsistent motion trajectories? How robust is SAW when the confidence metric (entropy or margin score) is unreliable? What happens if teacher predictions are largely incorrect (low pre-trained accuracy, e.g., Swin-T 65% case in Table 1)? A deeper analysis of these aspects would help clarify the method’s stability.

**Questions:**

The proofs in Section 3.2 and Appendix A.1–A.2 assume perfect or near-perfect teacher–student pairing and access to a validation set with clean labels for estimating weights. In practice, these conditions may not hold, especially for continuously moving agents without well-tracked sequences.

---

### Note · Authors · 2025-11-13

I have read and agree with the venue's withdrawal policy on behalf of myself and my co-authors.